# Detection of AFB_1_ by Immunochromatographic Test Strips Based on Double-Probe Signal Amplification with Nanobody and Biotin–Streptavidin System

**DOI:** 10.3390/foods13213396

**Published:** 2024-10-25

**Authors:** Yifan Li, Zhenfeng Li, Baozhu Jia, Zhui Tu, Juntao Zeng, Jiarui Pang, Wenjie Ren, Zhibing Huang, Baoshan He, Zhihua Wang

**Affiliations:** 1State Key Laboratory of Food Science and Technology, Nanchang University, Nanchang 330047, China; tuzhui@ncu.edu.cn (Z.T.); hzbchem@163.com (Z.H.); 2National Engineering Research Center of Wheat and Corn Further Processing, Henan University of Technology, Zhengzhou 450001, China; liyifan6225@163.com (Y.L.); zengjuntao2004@163.com (J.Z.); jiarui_pang@stu.scau.edu.cn (J.P.); hebaoshan@126.com (B.H.); wangzhihua202202@163.com (Z.W.); 3Guangzhou Nabo Antibody Technology Co., Ltd., Guangzhou 510530, China; lizhenfeng306@126.com; 4College of Biology and Food Engineering, Guangdong University of Education, Guangzhou 510303, China; jbzjbz130@163.com

**Keywords:** immunochromatographic test strip, AFB_1_, nanobody, signal amplification

## Abstract

Aflatoxin B_1_ (AFB_1_) is highly toxic and difficult to prevent. It is mainly produced by fungi and exists in plants and animals and is classified by the World Health Organization as a class I carcinogen, posing a serious threat to human and animal health. Therefore, it is important to establish an efficient, sensitive, and on-site detection method for AFB_1_ to protect human health. The immunochromatographic test strip method is simple, sensitive, and can achieve real-time detection. However, traditional immunochromatographic test strips have low sensitivity due to their relatively weak optical properties. In this study, Nb-G8 was biotinylated using a chemical method. Two sizes of gold nanoflowers (AuNFs) were prepared and combined with biotinylated G8 and streptavidin to form two types of probes. These probes were sprayed on gold standard pads and expanded pads, respectively, to enhance the signals through the high affinity interaction between streptavidin and biotin. Under the optimal experimental conditions, the half maximal inhibitory concentration (IC_50_) of this method was 5.0 ng/mL and the limit of detection (IC_10_) was 0.03 ng/mL, which increased the sensitivity of the test strip by four-fold compared with that of the traditional biotinylated nanoantibody immunochromatography test strip and had a wider detection range. In conclusion, the use of a high-affinity amplification signal between biotin and streptavidin is a valuable method for the detection of aflatoxin.

## 1. Introduction

Aflatoxin, a secondary metabolite synthesized by the fungi *Aspergillus flavus* and *Aspergillus parasiticus*, encompasses over 20 distinct types. Among these, aflatoxin B_1_ (AFB_1_) is the most common and exhibits the highest toxicity. It is mainly found in agricultural products such as peanuts, maize, and walnuts [1,2]. AFB_1_ has strong thermal stability, making it difficult to degrade by ordinary cooking and heating. It possesses properties of teratogenesis, carcinogenicity, and mutagenesis. Consuming food contaminated with AFB_1_ over a long period can be detrimental to both human and animal health [3]. Consequently, the International Agency for Research on Cancer (IARC) has designated AFB_1_ as a Group 1 carcinogen [4,5]. In this regard, countries around the world have set strict limits for AFB_1_ content in food, and China has stipulated that the content of AFB_1_ in maize, maize flour, and maize products should not exceed 20 μg/kg [6], while the European Union has even more stringent regulations, requiring that the content of AFB_1_ in consumer products for human life should not exceed 2 μg/kg [7]. AFB_1_ can contaminate crops during their growth, harvest, transportation, and storage, which is a serious threat to human health, and its rapid detection is essential to ensure food safety [8]. Therefore, it is urgent to establish a rapid and accurate method for detecting AFB_1_ in the field.

Routine methods for determining AFB_1_ in food primarily include high-performance liquid chromatography (HPLC) [9], high-performance liquid chromatography/tandem mass spectrometry (HPLC-MS/MS) [9,10], and electrochemical analysis [11,12]. Liao [13] proposed a novel method to integrate nanobodies (Nbs) with a biomimetic mineralized metal–organic framework (MOF). This method encapsulated a large amount of succinylated horseradish peroxidase (sHRP) within the MOF and assembled Nbs on the MOF, catalytically amplifying the detection signal of 4-chloro-1-naphthol. Yang [14] developed a unique bursting agent for ochratoxin A (OTA) by introducing zeolitic imidazole frameworks (ZIFs). This transformed the classical polydopamine (PDA) bursting agent into a new structure, resulting in a novel bursting ECL immunosensor for OTA. These methods have high detection sensitivity and excellent quantitative analytical performance, but due to their complexity and expensive instrumentation, they are not appropriate for rapid on-site detection because of the high professionalism requirements for the operators [15]. Compared with the above methods, ELISA does not require complex pre-treatment and is currently the most commonly used enzyme immunoassay method [16,17]; however, there are still some limitations in the field detection of this method, and the detection accuracy needs to be improved [18]. The current immunochromatographic test strip technique is characterized by simplicity and rapidity and is considered to be a promising detection tool for on-site rapid detection [19,20].

Immunochromatographic test strip technology is simple and inexpensive, but detection sensitivity limits its application as a quantitative detection method [21]. Therefore, improvement of detection sensitivity has become a prerequisite for the rapid development of immunochromatographic strip technology [22], and various signal amplification strategies have been developed by domestic and foreign researchers [23,24]. Ren [25] used Prussian blue nanoparticles (PBNPs) as signal tags and nanoenzymes to bind horseradish peroxidase-labeled goat samples through electrostatic adsorption. Peroxidase-labeled goat anti-mouse IgG (GAMA-HRP) or GAMA was used as a probe coupled to a free monoclonal antibody (mAb). Since one mAb can bind multiple GAMA molecules at the same time, the indirect probe enables primary signal amplification, followed by secondary amplification through PBNP-catalyzed TMB chromatography, and finally tertiary amplification through binding to HRP. Zhou [26] proposed a polyethyleneimine (PEI)-assisted copper in situ growth (CISG) signal amplification strategy, in which prepared PEI@Cu^2+^ was adsorbed on the surface of gold nanoparticle probes, and under the action of a mild reducing agent, the copper shells were grown and deposited on the surface of the gold nanoparticle probes on the T-line, which improved the gold nanoparticle lateral flow sensor (AuNP-LFS) sensitivity. Chen [24] developed rapid detection methods for T-2 mycotoxin using dual AuNP signal amplification (DuoICS) and SeNPs (Se-ICS). The sensitivities of these methods were 1 ng/mL and 0.25 ng/mL, respectively, which are 3 and 15 times greater than the sensitivity of the traditional ICS method. These methods improve detection sensitivity through signal amplification but are more complex to operate. Biotin–streptavidin is also a good signal amplification system [27], and its amplification principle is that the biotinylated labeled probes can achieve multistage amplification of detection signals through the high specific affinity between Bio-SA without the need to use complex chemical cross-linking agents and under the condition of maintaining the original biological activity [28]. Nadezhda A [29] utilized the interaction between Bio-SA and gold-standard antibody, GNP-Bio and GNP-SA to form a trimer at the T-line to form a nanoparticle network structure for signal amplification. The detection limits (LODs) for cTnI and cTnT were 0.9 ng/mL and 0.4 ng/mL, respectively, which are three times lower than those of more widely used systems. Additionally, the concentration of the visual load was ten times lower.

Antibodies are the most widely used recognition elements in biosensors. Among these, nanobodies possess unique properties like small size, easy genetic modification, and high stability, which traditional antibodies lack. Consequently, nanobodies have replaced monoclonal antibodies as novel recognition elements in immunosensors and are widely used in various fields [30,31,32]. Yan [33] designed a biotin–streptavidin-amplified enzyme-linked immunosorbent assay (BA-ELISA) for detecting AFB_1_ in cereals. This method utilizes biotinylated nanobody Nb26 and streptavidin conjugated with polymerized horseradish peroxidase (SA-PolyHRP). This assay demonstrated a 3.6-fold increase in sensitivity and significantly reduced the amount of Nb required compared to the classical Nb26-based ELISA. Zhang [34] created an enzyme-linked amplification immunoassay (ECAIA) for detecting OTA in coffee. This assay employs a nanobody-alkaline phosphatase fusion protein (mNb-AP) and MnO_2_ nanosheets. The bifunctional mNb-AP specifically binds to OTA and converts ascorbic acid-2-phosphate (AAP) into ascorbic acid (AA). The oxidase-mimicking MnO_2_ nanosheets were reduced to Mn^2+^ by AA, catalyzing the generation of blue oxidation products by TMB for quantitative detection, with a limit of detection (LOD) of 3.38 ng/mL and an IC_50_ of 7.65 ng/mL.

In order to improve the optical characteristics of signal labels and enhance the detection sensitivity of immunochromatographic test strips (ICTSs), this study constructed a double-probe signal amplification immunochromatographic test strip for AFB_1_ detection based on the nanobody and biotin–streptavidin system. The nanoantibody G8 was biotinylated using chemical methods. Two sizes of AuNFs were prepared and combined with biotinylated G8 and streptavidin to form two types of probes. These probes were sprayed on gold standard pads and expanded pads, respectively. The principle of signal amplification was based on the high affinity between biotin and streptavidin, which effectively reduces the amount of the gold-labeled probes and improves the detection sensitivity of the test strip.

## 2. Materials and Methods

### 2.1. Experimental Reagents and Instruments

The reagents and apparatus are in Appendix A.

### 2.2. Expression and Purification of Nanobody G8

Extracted pET25b-G8 plasmids were transformed into *E. coli BL21* receptor cells. The specific experimental steps are shown in Appendix A.

### 2.3. Establishment of Indirect ELISA Based on Nanobody G8

Detailed steps are in Appendix A.

### 2.4. Chemical Biotin-Labeled Nanobodies

NHS–biotin was dissolved in dimethyl sulfoxide (DMSO) to a final concentration of 10 mM; BNHS solution was added to the prepared nanoantibody G8 so that the molar ratios were 1:10, 1:20, 1:30, and 1:40, and the mixture was incubated at room temperature with gentle agitation for 30 min. The resulting solution was dialyzed in 0.1 M PBS at 4 °C for 24 h and changed every 8 h. The biotin-labeled G8 nanoantibody was obtained and stored at −20 °C.

### 2.5. Establishment of Labeled Nanobody BA-ELISA Based on Chemical Biotin Method

The artificial antigen AFB_1_-BSA was encapsulated and sealed as in Section 2.4. Add 50 μL of G8-Biotin with different molar ratios and 50 μL of AFB_1_ standards with different concentrations (0, 0.01, 0.1, 1, 5, 10, 50, 100, 200 ng/mL). The remaining steps are the same as Section 2.3 after diluting the nanoantibody.

### 2.6. Preparation of Gold Nanoflowers with Two Particle Sizes

According to the literature [35], the addition of seed liquid can significantly regulate the particle size of gold nanoparticles, and reducing the addition of seed liquid can increase the particle size of gold nanoparticles; the addition of HAuCl_4_ does not have a significant effect on the particle size, but it can significantly regulate the morphology of the gold nanoflowers, and reducing its addition is favorable for the generation of AuNFs. Therefore, AuNFs of different particle sizes were prepared by varying the amounts of HAuCl_4_ and seed solution added.

The seed solution was prepared using the trisodium citrate reduction method [35]. Detailed steps are in Appendix A. Gold nanoflowers were prepared using the seed growth method [36]. Detailed steps are in Appendix A.

### 2.7. Preparation of G8-Biotin Nanobodies and Streptavidin-Labeled AuNF Probes

The gold nanoflower solution was adjusted to the optimal pH with 1% K_2_CO_3_, and the appropriate amount of G8-Biotin nanoantibody was added to the gold nanoflower solution drop by drop, followed by magnetic stirring for 30 min. Then, 100 μL of 10% Bovine Serum Albumin (BSA) was added, and stirring continued for 15 min. The labeled colloidal gold solution was centrifuged at 7000 r/min for 10 min at 4 °C, the supernatant was discarded, and the precipitated gold-labeled probes were removed. The precipitated gold-labeled probes were resuspended with gold-labeled probe resuspension solution and stored at 4 °C. The preparation method of the streptavidin-labeled AuNF probe was the same as the method described above.

### 2.8. Preparation of Biotin-BSA

Ten mg of BSA was dissolved in 2 mL of PBS. Then, 10 mM of BNHS solution was added to achieve a molar ratio of 1:15 with gentle shaking. The reaction mixture was incubated at indoor temperature for 45 min and then dialyzed with 0.1 M PBS at 4 °C for 24 h to remove unreacted biotin.

### 2.9. Assembly of Test Strips

Different concentrations of AFB_1_-BSA and biotin–BSA were sprayed onto the nitrocellulose membrane as the detection line and quality control line, respectively, using a membrane scribing and gold spraying integrated machine. The scribed nitrocellulose (NC) membrane was placed in a 37 °C vacuum drying oven and dried overnight. The gold standard pad, expanding pad and sample pad were cut to a width of 1.0 cm, and the absorbent pad was cut to a width of 1.75 cm for later use. Before assembling the test strips, the sample pads, gold standard pads, and expanding pads were soaked in gold standard pad treatment solution (6% trehalose, 1% BSA, 0.5% Tetranc1307 dissolved in 20 mM borate buffer solution) for 30 min and then dried in a vacuum drying oven at 37 °C. Finally, the nitrocellulose membrane, expanding pad, gold standard pad, sample pad, and absorbent pad were sequentially assembled on a PVC backing plate, ensuring an overlap of 2-3 mm between each pad. The assembly was then cut into 3 mm wide strips with a strip cutter and stored in a sealed bag for dry storage.

### 2.10. Optimization of Test Strip Parameters

The specific optimization steps are shown in Appendix A.

### 2.11. Performance Evaluation of Immunochromatographic Test Strips

#### 2.11.1. Establishment of Standard Curves for Immunochromatographic Test Strips

Under the optimal conditions, different concentrations of AFB_1_ standard solutions were prepared so that the final concentrations of 0, 0.01, 0.1, 1, 5, 10, 50, 100, and 200 ng/mL were dropwise added to the sample pad. OD_T_/OD_C_ values were determined using a chromatographic readout device, and three parallels were made for each sample. Detailed information is in Appendix A.

#### 2.11.2. Evaluation of Specificity

In order to study the specificity of the constructed test strips for the detection of AFB_1_, other toxins (T-2, OTA, DON, ZEN and AFM_1_) and blank samples were assayed using the strips to evaluate the specificity of the test strips.

### 2.12. Actual Sample Testing

To further assess the application value of the constructed biotinylated nanobody immunochromatographic test strips, negative corn samples were used for spiking experiments. The procedure for the maize samples is shown in Appendix A. These concentrations were quantified using a standard curve. The method was evaluated by assessing spiked recoveries and coefficients of variation.

## 3. Results and Discussion

### 3.1. Principle of Test Strip Detection

In this paper, two kinds of AuNFs with different particle sizes are combined, highly sensitive biotinylated nanoantibodies G8 and SA to prepare AuNFs@G8-Bio and AuNFs@SA dual probes to construct signal-amplifying immunochromatographic test strips for the rapid and sensitive detection of AFB_1_. The differing particle sizes of the dual probes cause variations in their chromatographic speeds on NC membranes; signal amplification was achieved by a highly stable and high-affinity biotin–streptavidin system. Gold nanoflowers with two particle sizes were prepared by varying the addition of HAuCl_4_ and gold seeds. The 65 nm AuNFs were coupled with G8-Bio by electrostatic adsorption as the gold standard probe, and the 110 nm AuNFs were coupled with SA by electrostatic action as the expanded probe (Figure 1A). The signal amplification test strips were composed of a sample pad, gold standard pad, expansion pad, NC membrane, and absorbent pad from left to right, and the gold standard probe and the expansion probe were sprayed on the gold standard pad and the expansion pad, respectively, and AFB_1_-BSA and Bio-BSA were sprayed at the T line and the C line, respectively, of the NC membrane (Figure 1B). The principle of double-probe signal amplification in immunochromatographic test strips is primarily based on the antigen–antibody indirect competitive reaction mechanism, as illustrated in Figure 1C. Fifty μL of buffer solution was added dropwise to the spiking port, and when there was no AFB_1_ in the sample, it flowed through the expanding pad and the gold standard pad under the force of the capillaries. The small particle size of AuNFs@G8-Bio resulted in a rapid flow rate. Initially, AuNFs@G8-Bio was bound to AFB_1_ on the T line. Subsequently, AuNFs@SA flowed through the T line and was captured there due to the high affinity between biotin and streptavidin, intensifying the blue color of the T line. The remaining probe was captured on the C line, leading to blue coloration on both the T and C lines. When AFB_1_ is present in the sample, AFB_1_ first binds to AuNFs@G8-Bio, which cannot be captured when flowing through the T line; thus, AuNFs@SA cannot be captured when flowing through the T line and the probe is captured when flowing through the C line, which does not show any coloration in the T line, and shows a blue color in the C line. Signal amplification was accomplished by the high affinity between biotin and streptavidin to accomplish the binding of the gold standard probe and the expanded probe on the T line. Less AuNFs@G8-Bio is used to achieve a higher gray value of T-line and improve the sensitivity of test strip detection.

### 3.2. Expression and Purification of Nanobody G8

The plasmid pET25b-G8 was transformed into receptor cells *E. coli* BL21 (DE3) and expressed for 8 h after 0.1 mM IPTG induction at 25 °C and 200 rpm. The expressed bacterial fluid was broken by an ultrasonic cell crusher, and the supernatant was taken for nickel column purification. Protein expression was analyzed using SDS-PAGE, and the results are displayed in Figure 2A. The purity of the target proteins eluted by 100 mM imidazole concentration was the highest, and the concentration was also the highest, and bands with a size of about 18 KDa appeared, which indicated that the nanoantibody G8 could be solubilized to be expressed in Escherichia coli after induced.

### 3.3. Establishment of Standard Curve for Indirect Competitive ELISA of Nanobody G8

The working concentration of Nb-G8 was determined at an encapsulated concentration of 2 μg/mL of AFB_1_-BSA, and an indirect competitive ELISA curve for Nb-G8 was established. As shown in Figure 2A, Logistic equations were used to fit the ELISA inhibition curves of AFB_1_-BSA versus AFB_1_ competing for G8. The equation is given as follows:(1)$$y=A_{2}+(A_{1}-A_{2})/[1+{\left(\frac{X}{X_{0}}\right)}^{P}]$$

The horizontal coordinate is the concentration of AFB_1_ and the vertical coordinate is the binding ratio B/B_0_ × 100% (B represents positive OD_450_ and B_0_ represents negative OD_450_). The fitted standard curve equation is as follows:(2)$$y=3.312+93.986/(1+0.049X^{0.617})$$

The IC_50_ was calculated to be 20.86 ng/mL.

### 3.4. Establishment of Standard Curve for Indirect Competitive ELISA of Nb-G8 Labeled by Chemical Biotin Method

The half maximal inhibitory concentration (IC_50_) and limit of detection (IC_10_, LOD) were defined as the target concentrations that produce 50% and 10% inhibition of the signal (B/B_0_), respectively. Biotin binding to streptavidin with high specificity and good stability is a hotspot in the field of bioassays and can play the role of signal amplification. Among them, the preparation of biotinylated antibodies is a very important part. In general, multiple biotin can be bound to a single antibody, each of which can be coupled to SA-HRP, thereby increasing the signal intensity. In this study, biotinylated nanoantibodies were prepared by chemical modification using four different biotin molar ratios for labeling Nb-G8, which allowed the streptavidin to be coupled to the biotin without affecting the specific recognition of AFB_1_-BSA by the nanoantibodies. The biotinylated nanobodies were prepared with the molar ratios of nanobodies to biotin of 1:10, 1:20, 1:30, and 1:40, and BA-ELISA was constructed to evaluate the effect of biotinylation, with IC_50_ as the evaluation standard. The results are shown in Figure 2B; the highest sensitivity was observed when the molar ratio was 1:20 and the IC_50_ was 10.42 ng/mL, but the IC_50_ value increased gradually with the increase in the molar ratio, which might be an excess of biotin that occupied the binding site with the antigen. Compared with the conventional Nb-ELISA, biotinylated Nb-G8 had good binding activity and sensitivity to AFB_1_, and under the same experimental conditions, the biotin–streptavidin-based BA-ELISA had higher sensitivity than the Nb-ELISA.

### 3.5. Characterization of AuNFs with Different Particle Sizes

AuNFs are distributed with radioactive dendrites on the surface and thus have enhanced optical properties, a larger specific surface area, and stronger binding affinity compared to conventional AuNSs, increasing the contact area for immunoreactivity. The addition of seed solution and HAuCl_4_ is the main factor affecting the particle size, and in this experiment, different particle sizes of AuNFs were synthesized by adjusting the dosage. Two particle sizes of AuNFs, physical images, transmission electron microscopy scans, and particle size analysis are shown in Figure 3. By changing the addition of seed liquid and HAuCl_4_, it can be clearly seen that the AuNFs have a significant difference in particle size and color and all have large, spiky, convex, and well-dispersed uniform particles. When 375 μL of HAuCl_4_ and 2 mL of seed solution were added, the color was mauve, the particle size was small, and the spinous convexity was obvious (Figure 3A). When 375 μL and 0.5 mL of seed solution were added, the color was blue, the particle size was larger, and the spiny convexity was obvious. The TEM images of the two particle sizes of AuNFs were analyzed using the software Nano Measurer 1.2.5. As can be seen in Figure 3C,D, the small-sized AuNFs are around 67 nm and the large-sized AuNFs are around 110 nm, and these results indicate that the two particle sizes of AuNFs were successfully prepared and the particle sizes are as expected.

### 3.6. Optimization of Detection Conditions for Immunochromatographic Test Strips

#### 3.6.1. Optimization of pH for Preparation of AuNFs@anti-Nb-G8-Bio Probes

In the preparation of gold-labeled probe AuNFs@anti-Nb-G8-Bio, the nanobody G8-Bio was coupled with AuNFs by electrostatic adsorption, in which pH is the main reason affecting the coupling efficiency; therefore, in this experiment, 1% K_2_CO_3_ solution was used to adjust the pH to 5.5, 6.0, 6.5, and 7.0, respectively. The results are presented in Figure 4A. When the solution pH is 5.5, the T line shows almost no color; when the solution pH is 6.0, the T line has the highest color intensity and is the same color as the C line. When the solution pH is 6.5 or 7.0, both the T and C lines show color, but the T line has a relatively weaker color intensity. Therefore, a pH of 6.0 was selected as the optimal condition.

#### 3.6.2. Optimization of the Amount of Nanobody for the Preparation of AuNFs@anti-Nb-G8-Bio Probes

In the preparation of the gold-labeled probe AuNFs@anti-Nb-G8-Bio, the amount of nanoantibody affects the stability of the probe and the sensitivity of the test strip. Only when the amount of nanoantibody added is optimal can the dead gold phenomenon be avoided during the preparation of the probe and ensure that the probe is always stable. If the amount of antibody is excessive, it will lead to antibody wastage and negatively impact the sensitivity and minimum detection limit of the test strip. Therefore, during the optimization process, it is crucial to minimize the amount of nanoantibody added. This adjustment aims to enhance the sensitivity of the test strips while ensuring that the probes remain stable and that the T and C lines display normal and consistent colors. As shown in Figure 4B, when the amount of antibody added was 10 μL, the dead gold phenomenon occurred in the probe, resulting in the T line not showing color. When 15, 20, and 25 μL of antibody was added, the probe performance remained stable, and the T-line color intensity showed little variation. The T-line color intensity was relatively higher at 15 μL of antibody. Therefore, considering both cost and sensitivity, 15 μL of nanobody was selected as the optimal condition.

#### 3.6.3. Preparation of AuNFs@SA Probe pH Optimization

The solution pH is a crucial factor influencing the adsorption efficiency of AuNFs with SA. K_2_CO_3_ (1%) was used to adjust the solution pH, as shown in Figure 4C. pH was increased from left to right in the order of 5.5, 6.0, 6.5, and 7.0. The color intensity of the T line was maximum when the pH was 6.5. When the pH value is greater than or less than 6.5, the color intensity of the T-line decreases. When the color intensity of the T-line is the greatest, it indicates that the AuNFs@SA binds better to the AuNFs@anti-Nb-G8-Bio on the T-line. That is, AuNFs adsorbed SA with the highest efficiency. Therefore, a solution pH of 6.5 was chosen as the optimum condition.

#### 3.6.4. Optimization of SA Concentration for Preparation of AuNFs@SA Probes

At the optimal solution pH, varying concentrations of SA were added to the AuNF solution with small particle sizes, as shown in Figure 4D. As the SA concentration increased, the T-line color intensity gradually increased. When the SA concentration reached 0.6 mg/mL, the T-line color intensity peaked and then remained stable with further increases in SA concentration. Therefore, an SA concentration of 0.6 mg/mL was selected as the optimal concentration.

#### 3.6.5. Optimization of Expanded Pad Positions

The placement of AuNFs@SA as a signal amplification probe directly affects the performance of signal amplification. The nanobodies were coupled with small-sized AuNFs, and SA was coupled with large-sized AuNFs, which resulted in different crawling velocities of the two probes in the NC membrane due to the different sizes of the AuNF particles. The large particle size makes it climb slower on the NC membrane, and the small particle size makes it climb faster on the NC membrane. As shown in Figure 5A, this experiment optimizes the positional situation of the gold standard pad and the expansion pad: lane 1 from top to bottom for gold standard and expanded pads and lane 2 from top to bottom for expanded and gold standard pads. The results confirmed that the position of the two pads had an effect on signal amplification; when the gold standard pad was on top, the T and C lines were well colored, and the AuNFs@SA probe acted as a signal amplifier. When the expanding pad was on top, the color intensity of the T line was weak and that of the C line was strong, indicating that the crawling speed of the AuNFs@anti-Nb-G8-Bio probe does not make up for the difference in the distance between the two pads and indicating that the AuNFs@SA probe takes the lead in flowing through the T line, which does not act as a signal amplifier. Therefore, in this experiment, the gold standard pad was placed on top and the amplification pad was placed on bottom.

#### 3.6.6. Optimization of the Ratio of AuNFs@anti-G8-Bio to AuNFs@SA

The amount of AuNFs@SA used as a signal amplification probe is one of the main reasons affecting the signal amplification effect and the sensitivity of the test strip. As shown in Figure 5B, from left to right, the ratios of AuNFs@anti-G8-Bio to AuNFs@SA probes were 2:1, 1:1, 2:3, 1:2, and the spraying amount of AuNFs@anti-G8-Bio was 7 μL/cm, and the color intensity of the T-line was gradually enhanced with the increase in the spraying amount of the AuNFs@SA probe, and then the T-line color intensity tended to be stable at 2:3. Therefore, the optimal ratio of probe AuNFs@anti-G8-Bio to AuNFs@SA is 2:3, with the AuNFs@SA probe sprayed at 11 μL/cm and the AuNFs@SA probe sprayed at 10% more to ensure good signal amplification.

#### 3.6.7. T and C Line Concentration Optimization

The concentration of artificial antigen encapsulated on the T-line is another important factor affecting the sensitivity and accuracy of the test strip. Too high a concentration of encapsulation results in too many probes being captured on the T-wire, which leads to a decrease in the degree of response of the T-wire when detecting positive samples, resulting in a decrease in the sensitivity of the test strip. Too low a concentration of encapsulation will prevent the T-line from capturing the probe, leading to false-positive results and affecting the accuracy of the test strip. Therefore, this experiment optimized the concentration of AFB_1_-BSA encapsulated on the T-line. As illustrated in Figure 6A, the T-line’s color intensity increased progressively with higher AFB_1_-BSA concentrations. When the AFB_1_-BSA concentration reached 200 μg/mL, the increase in T-line color intensity plateaued. Considering the sensitivity and accuracy of the test strips, the optimal AFB_1_-BSA concentration was determined to be 200 μg/mL. To ensure that the color intensity of the C-line matched that of the T-line, achieving a T/C ratio close to 1, the optimal concentration of Bio-BSA in the C-line was found to be 0.3 mg/mL (Figure 6B).

### 3.7. Evaluation of Immunochromatographic Test Strip Assays

#### 3.7.1. Establishment of Standard Curves and Sensitivities

The detection sensitivity of the test strips was further evaluated under optimal experimental conditions. Different concentrations of AFB_1_ standard solutions (0, 0.01, 0.1, 1, 5, 10, 50, 100, and 200 ng/mL) were prepared, the logarithm of which was taken as the horizontal coordinate and B/B_0_ × 100% as the vertical coordinate to construct the competitive inhibition curves of the test strips. In order to reduce the systematic error, three sets of independent measurements were carried out for each concentration during the assay. The results are shown in Figure 7A. The grey value of T/C of the test strip was read using a chromatographic reader and calculated by Logistic equation as follows:(3)$$y=-1.688+103.123/(1+0.197X^{0.418})$$

The IC_10_ was calculated to be 0.03 ng/mL, the IC_50_ was 5.0 ng/mL, and the detection range (IC_20_~IC_80_) was 0.2 ng/mL~119.4 ng/mL. The conventional ICTS without signal amplification (Figure 7B) was calculated to have an IC_10_ of 0.12 ng/mL, the IC_50_ was 9.8 ng/mL, and the detection range (IC_20_~IC_80_) was 0.76 ng/mL~85.26 ng/mL. Of particular note, the dual-probe signal-amplified ICTS provides a four-fold increase in signal enhancement sensitivity with a wider detection range compared to the unamplified signal conventional ICTS.

#### 3.7.2. Evaluation of the Specificity of Immunochromatographic Test Strips

To further validate the specificity of the immunochromatographic test strip, eight common fungal toxins, OTA, ZEN, DON, T-2, AFM_1_, AFB_2_, AFG_1_, and KAN, were selected as interfering toxins for validation, and a positive control was set as the standard solution of AFB_1_, and a blank control was set as the sampling buffer. The above substances were diluted to 50 ng/mL, and 50 μL of each substance was taken for the spotting, which was repeated three times, and the T-line gray value was determined using an immunochromatographic. The results are shown in Figure 8. Five interfering toxins in high-concentration condition test-strip T-line color intensity compared to the blank value showed almost no change. The AFB_1_ standard spot sample test-strip T-line color intensity compared to the blank value decreased significantly, indicating that the immunochromatographic test strips constructed in this experiment and the other five fungal toxins do not have cross-reactions and have a very strong specificity for AFB_1_.

### 3.8. Actual Sample Testing

In order to investigate its feasibility for the detection of AFB_1_ in real samples, maize was selected as a real sample and spiked in this study, and the accuracy and precision was evaluated by assessing both the recovery and the coefficient of variation. Unspiked maize samples were tested for the non-detection of AFB_1_. The results of the spiking experiments are shown in Table 1, and the spiked recoveries of the test strips ranged from 100~102%, and the coefficients of variation ranged from 0.98~10.0%, which indicated that the accuracy and precision of the constructed ICTSs could be used for the detection of AFB_1_. The key parameters for detecting AFB_1_ using ELISA, HPLC, and electrochemical immunosensors are shown in Table 2. In comparison, the detection limit of the method used in this experiment is not significantly different, but the time required is shorter.

## 4. Conclusions

In summary, this study successfully expressed and purified G8 nanoantibodies and prepared biotinylated nanoantibodies by the chemical biotin method and established a nanobody-based dual-probe signal amplification immunochromatographic test strip for high-sensitivity detection of AFB_1_. By electrostatic adsorption, 67 nm particle-size AuNFs were coupled with nanobody G8-Bio as a capture probe, and 110 nm AuNFs were coupled with SA as a signal amplification probe, sprayed on gold standard pads and expanding pads, respectively. The AuNFs@G8-Bio probe, owing to the small particle size of the gold flower and the position of the gold standard pad on top, ascended faster on the NC membrane than the AuNFs@SA probe and was captured by the T-line first. Subsequently, when the AuNFs@SA flowed through the T-line, the high affinity between biotin and streptavidin enabled it to be recaptured by the AuNFs@SA already immobilized on the T-line. This process amplified the signal and enhanced the sensitivity. The method demonstrated a lowest detection limit (IC_10_) of 0.03 ng/mL, with an IC_50_ of 5.0 ng/mL. The detection range, spanning from IC_20_ to IC_80_, was between 0.2 ng/mL and 119.4 ng/mL. The detection limit of the commercially available brand Feice Biological test strip is 1 ug/kg, compared to the minimum detection limit of 0.03 ug/kg in this experiment, which is more than 30 times more sensitive than the commercially available test strip. Compared with the traditional nanobody G8-Bio immunochromatographic strips based on the nanobody G8-Bio, the detection range was significantly expanded, and its sensitivity was improved by four-fold; compared with the BA-ELIAS and the traditional ic-ELISA, the IC_50_ of the test strip was reduced by two-fold and four-fold, respectively. The test strip was also successfully applied to actual corn samples, demonstrating the practicality and effectiveness of the developed signal amplification immunochromatographic test strip in real-world applications. Therefore, the biotinylated nanoantibody-based dual-probe signal amplification immunochromatographic test strip developed in this paper is not only simple, fast, convenient, and low-cost but also has the advantages of higher sensitivity and accuracy, which can be used as a potential tool for the on-site detection of AFB_1_.

## Figures and Tables

**Figure 1 foods-13-03396-f001:**
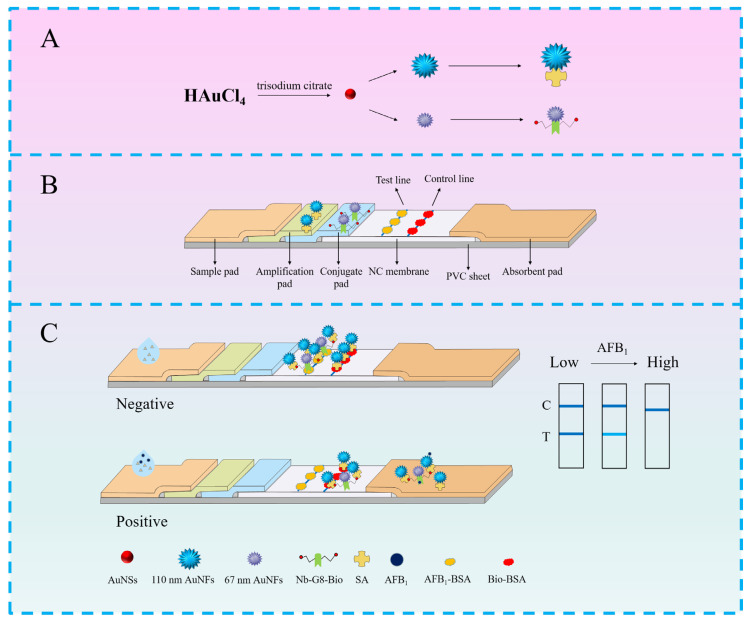
(**A**) Schematic diagram of the preparation process of AuNFs@G8-Bio and AuNFs@SA. (**B**) Structure of immunochromatographic test strips based on double-probe signal amplification by nanoantibody and biotin–streptavidin system and (**C**) principle of detection of AFB_1_ with test strips.

**Figure 2 foods-13-03396-f002:**
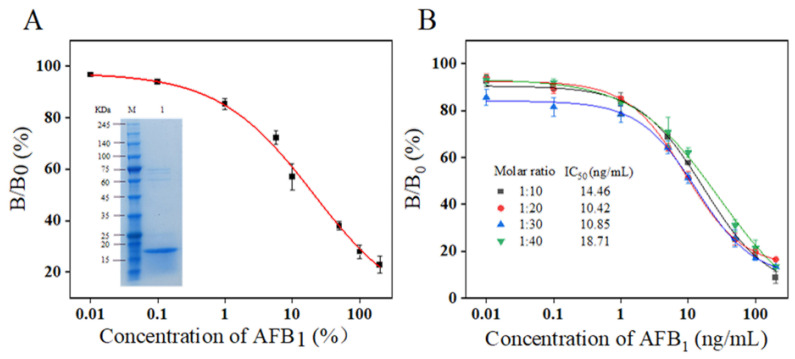
Nb-G8 indirect competition ELISA standard curve and SDS-PAGE analysis: (**A**) lane M: protein marker; lane 1: 100 mM imidazole eluent. (**B**) Nb-G8-Bio-based BA-ELISA standard curve.

**Figure 3 foods-13-03396-f003:**
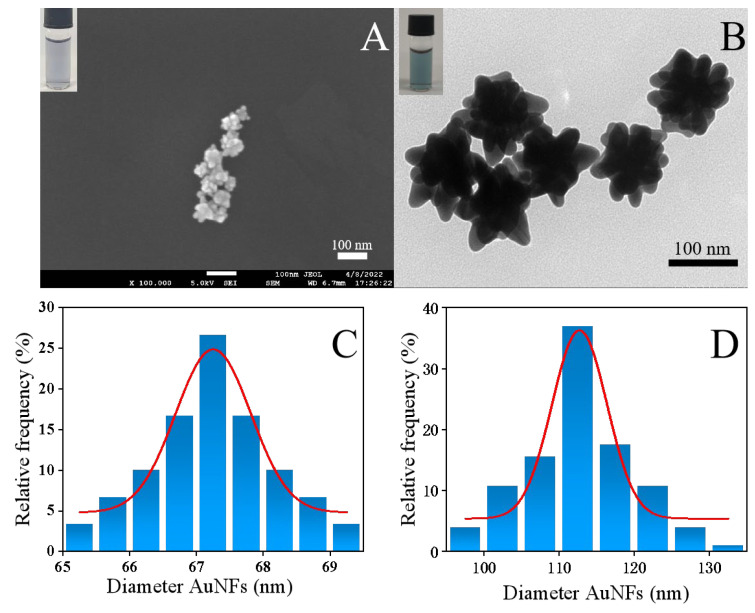
Characterization of two particle sizes of AuNFs. (**A**,**B**) Physical images and TEM scans of AuNFs of two particle sizes. (**C**,**D**) Histograms and normal distribution curves of the particle size distribution of AuNFs of two particle sizes.

**Figure 4 foods-13-03396-f004:**
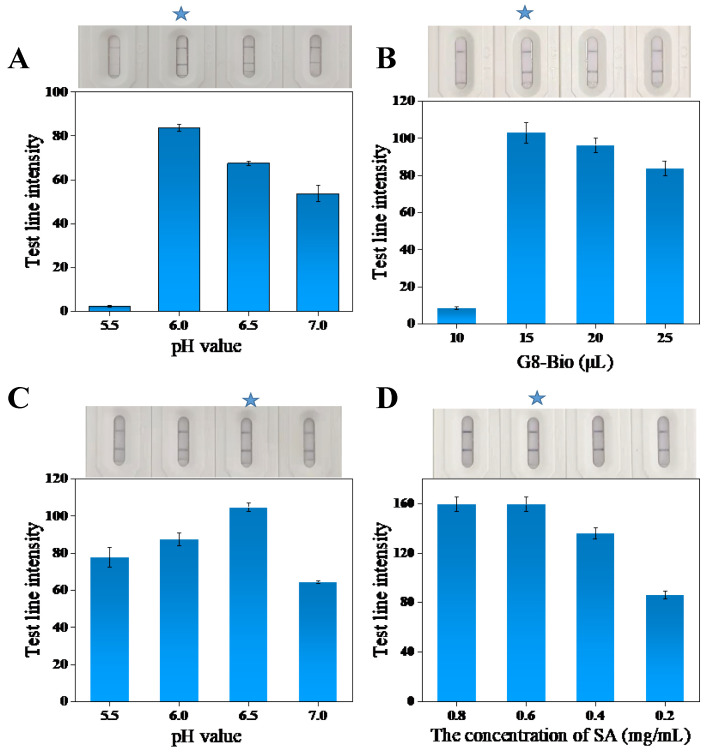
Signaling probe pH optimization (**A**) and G8-Bio amount optimization (**B**); expanded probe pH optimization (**C**) and SA concentration optimization (**D**). (The star marking is the optimal condition).

**Figure 5 foods-13-03396-f005:**
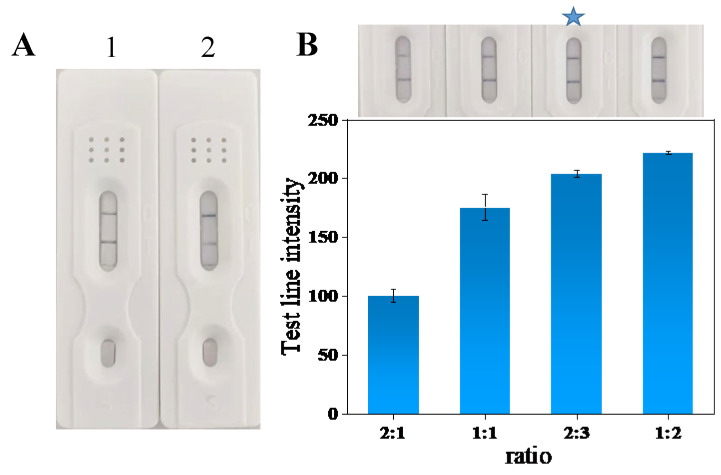
Optimization of gold standard probe and expanded probe positions, where 1 is the gold standard pad on top and 2 is the expanded pad on top (**A**), and ratio optimization (**B**). (The star marking is the optimal condition).

**Figure 6 foods-13-03396-f006:**
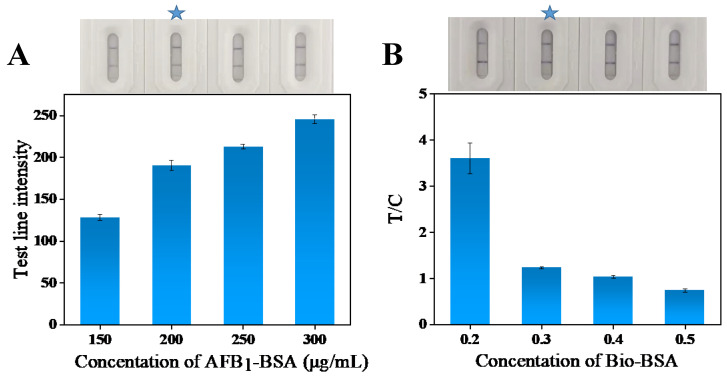
Optimization of AFB_1_-BSA concentration in line T (**A**) and Bio-BSA concentration in line C (**B**). (The star marking is the optimal condition).

**Figure 7 foods-13-03396-f007:**
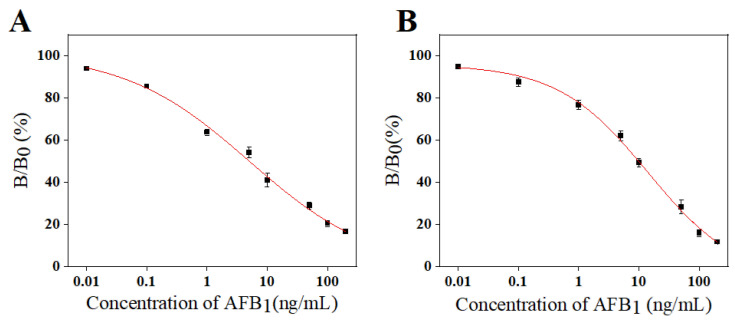
Dual-probe signal amplification test strip standard curve (**A**) and conventional test strip standard curve (**B**).

**Figure 8 foods-13-03396-f008:**
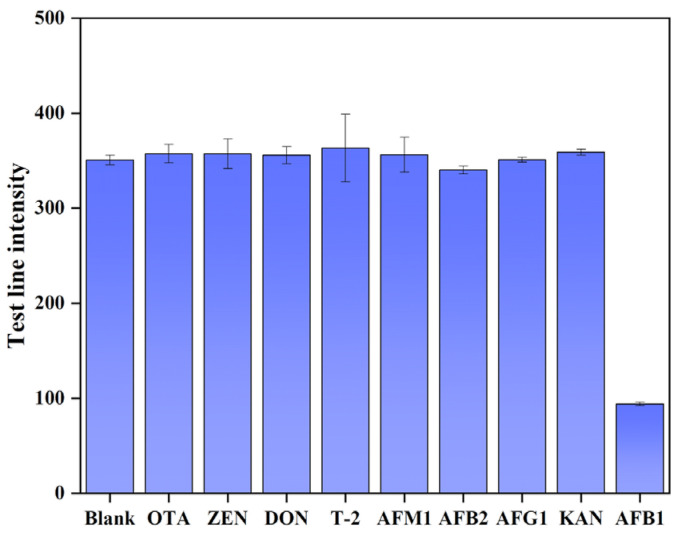
Specificity evaluation.

**Table 1 foods-13-03396-t001:** Spiked recovery of AFB_1_ in maize samples.

Actual Sample	Mycotoxin	Additive Quantity (ng/mL)	Measured Quantity (ng/mL)	Recovery Rate (%)	Coefficient of Variation (%)
Sorghum	AFB_1_	0	ND	ND	ND
		0.1	0.1 ± 0.01	100	10.0
		1.0	1.0 ± 0.08	100	8.0
		10	10.20 ± 0.10	102	0.98

**Table 2 foods-13-03396-t002:** Analysis performance of different AFB_1_ detection methods.

Method	Sensitivity	Detection Line (LOD)	Linear Detection Range	Analysis Time
ELISA	secondary	0.01–0.1 µg/kg	0.02–100 ng/mL	1–2 h
HPLC	high	0.01–0.05 µg/kg	0.1–50 ng/mL	30–60 min
Electrochemical immunosensor	high	0.001–0.01 µg/kg	0.001–50 ng/mL	10–30 min

## Data Availability

The original contributions presented in the study are included in the article; further inquiries can be directed to the corresponding author.

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
