# Peer review of "Detection of AFB1 by Immunochromatographic Test Strips Based on Double-Probe Signal Amplification with Nanobody and Biotin–Streptavidin System"

_foods, 2024, doi:10.3390/foods13213396_

Round 1
Reviewer 1 Report
Comments and Suggestions for Authors
General Comments: This research explores the development of a highly sensitive lateral flow assay (LFA) platform aimed at detecting Aflatoxin B1 (AFB1), a potent carcinogen hazardous to both human and animal health. The study utilizes key materials such as biotinylated Nb-G8 antibodies and engineered gold nanoflower probes to enhance signal detection by leveraging the strong interaction between streptavidin and biotin. Despite its promising approach, several experimental flaws and logical inconsistencies have been identified. It is recommended that these major issues be addressed and resolved.
Major Issues:
1. Limit of Detection (LoD):
The LoD for this platform has not been calculated. To accurately evaluate the sensitivity of the proposed method, a comparison with commercially available AFB1 detection kits (e.g., Attogene AU2051) is essential. Such a comparison would provide a clearer perspective on the platform's performance and its sensitivity relative to existing solutions.
2. Summary Table of Analytical Performance:
A summary table should be included to compare the analytical performance of various AFB1 detection methods such as ELISA, HPLC, and electrochemical immunosensors. The table should provide key parameters including sensitivity, limit of detection, linear detection range, and analysis time, allowing for a comprehensive comparison of the proposed platform with other established methods.
3. Specificity Evaluation:
The specificity assessment presented in this study is inadequate due to the absence of control antigens. To enhance the evaluation, additional control substances such as aflatoxins B2 (AFB2), aflatoxins G1 (AFG1), sulfadimethoxine (SDM), lead ions (Pb²⁺), and kanamycin (KAN) should be included. These controls will ensure a more thorough assessment of cross-reactivity and improve the accuracy of the assay in distinguishing AFB1 from other potentially interfering substances.
Author Response
Comments 1: Limit of Detection (LoD):
The LoD for this platform has not been calculated. To accurately evaluate the sensitivity of the proposed method, a comparison with commercially available AFB1 detection kits (e.g., Attogene AU2051) is essential. Such a comparison would provide a clearer perspective on the platform's performance and its sensitivity relative to existing solutions.
Response 1: Thank you for pointing it out. We agree with your point of view and have added a commercially available reagent kit in line 452 for comparison.
Comments 2: Summary Table of Analytical Performance:
A summary table should be included to compare the analytical performance of various AFB1 detection methods such as ELISA, HPLC, and electrochemical immunosensors. The table should provide key parameters including sensitivity, limit of detection, linear detection range, and analysis time, allowing for a comprehensive comparison of the proposed platform with other established methods.
Response 2: Thank you for pointing it out. We agree with your point of view and have listed Table 2 in the article to analyze the performance of different detection methods.
Comments 3: Specificity Evaluation:
The specificity assessment presented in this study is inadequate due to the absence of control antigens. To enhance the evaluation, additional control substances such as aflatoxins B2 (AFB2), aflatoxins G1 (AFG1), sulfadimethoxine (SDM), lead ions (Pb²⁺), and kanamycin (KAN) should be included. These controls will ensure a more thorough assessment of cross-reactivity and improve the accuracy of the assay in distinguishing AFB1 from other potentially interfering substances.
Response 3: Thank you for your question. We agree with your point of view and have added an evaluation of specificity.
Reviewer 2 Report
Comments and Suggestions for Authors
I review the manuscript: “Detection of AFB1 by immunochromatographic test strips based on double probe signal amplification with nanobody and biotin-streptavidin system” (foods-3248535. The idea is clearly presented but results are described (not discussed), additionally, it requires a deeper description of strip construction.
Some comments:
a) The article cited 35 references, 33 for introduction and 2 in experimental section (for AuNPs synthesis). Discussion was performed without any reference support. Results are just described and there is not a clear discussion.
b) Line 72, please define PBNPs.
c) Experimental section is described in supplementary material. 2.9 must be schematized considering the strip as the fundamental part, it must include a clear description employed for strip construction.
d) The Strip analytical performance was evaluated; however, it must be included the protocol used for validation (including a reference). This point is critical for comparison of LOD.
e) Figure 1 can be an excellent scheme. However, it is not clear enough to understand the immunochromatographic test strips detection mechanism. It would be useful to highlight the AFB1 symbol considering it is the analyte.
f) Please review the significant figures in Table 1. What about precision (inter-, intra-day) and accuracy. There is an official method for AFB1 analysis, then accuracy must be evaluated by comparison with this methodology (recovery tests are not enough). Additionally, it must be including the analysis of more samples (one sample + 3 standard additions) are not enough for validation.
Author Response
Comments 1: The article cited 35 references, 33 for introduction and 2 in experimental section (for AuNPs synthesis). Discussion was performed without any reference support. Results are just described and there is not a clear discussion.
Response 1: Thank you for pointing out the issue. We have added discussion content in the last paragraph.
Comments 2: Line 72, please define PBNPs.
Response 2: Thank you for pointing it out. We agree with your point of view and have defined PBNP on line 72.
Comments 3: Experimental section is described in supplementary material. 2.9 must be schematized considering the strip as the fundamental part, it must include a clear description employed for strip construction.
Response 3: Thank you for pointing out how the treatment solution was prepared in section 2.9.
Comments 4: The Strip analytical performance was evaluated; however, it must be included the protocol used for validation (including a reference). This point is critical for comparison of LOD.
Response 4: Thank you for pointing out the issue. We have added a definition of LOD when IC50 first appeared in section 3.4.
Comments 5: Figure 1 can be an excellent scheme. However, it is not clear enough to understand the immunochromatographic test strips detection mechanism. It would be useful to highlight the AFB1 symbol considering it is the analyte.
Response 5: Thank you for pointing out the issue. We have made modifications to Figure 1 and enlarged the AFB1 symbol.
Comments 6: Please review the significant figures in Table 1. What about precision (inter-, intra-day) and accuracy. There is an official method for AFB1 analysis, then accuracy must be evaluated by comparison with this methodology (recovery tests are not enough). Additionally, it must be including the analysis of more samples (one sample + 3 standard additions) are not enough for validation.
Response 6: Thank you for your question. The data in Table 1 shows daytime accuracy, and in Table 2, we have added some key parameters for detecting AFB1 using other methods.
Reviewer 3 Report
Comments and Suggestions for Authors
The article shows an interesting option for a sensitive detection of aflatoxin B1 in food samples, the authors carried out analysis of spiked samples with good recovery percentages, thus, the method seems to be promising for its application in food analysis. However, there are some indications that must be addressed to the manuscript.
- Line 15 authors say: Aflatoxins are found in plants ,crops and food products like peanuts, grains and nuts as contaminants from fungi, that is, vegetables and plants act like means of transmission of aflatoxins. But animals and people get infected through the ingestion of those products. It would be better if the drafting of this sentence is changed, because it implies that aflatoxins are produced by animals, rather than being produced by fungi.
- Line 32: The authors should put all the genus and species in italics. The authors should check all the text.
- Line 46: The authors should add the reference.
- Line 72: The authors should put the definition of PBNPs
- Line 112: The authors should define IC50 the first time it is mentioned.
- Line: despite DMSO being a common reagent, The authors should describe the definition.
- Line 114: The authors should define ICTS.
- Line 155: The authors should define BSA the first time it is mentioned aas same as all the other abbreviations used in the paper.
- Line 162: "10 mg of BSA was dissolved", The authors should correct the sentence to "were dissolved"
- Lines 153-160: The authors should rewrite this methodology in the past tense.
- Line 188: the subtitle could be changed to "Evaluation of specificity" to make it more clear.
- Line 345: the spelling of "optimisation" is incorrect. Also, the term optimization involves the use of a design of experiment, which was not made in this work. It could be more appropriate to replace this word with another term, such as "comparison" or "evaluation".
Author Response
Comments 1: Line 15 authors say: Aflatoxins are found in plants ,crops and food products like peanuts, grains and nuts as contaminants from fungi, that is, vegetables and plants act like means of transmission of aflatoxins. But animals and people get infected through the ingestion of those products. It would be better if the drafting of this sentence is changed, because it implies that aflatoxins are produced by animals, rather than being produced by fungi.
Response 1: Thank you for bringing up this point. We agree with your viewpoint and would like to revise the sentence to 'It is mainly produced by fungi and exists in plants and animals', and mark it in red in the original text.
Comments 2: Line 32: The authors should put all the genus and species in italics. The authors should check all the text.
Response 2: Thank you for pointing it out. We agree with your point of view and would like to change it to italicized text.
Comments 3: Line 46: The authors should add the reference.
Response 3: Thank you for pointing it out. We agree with your point of view and have added the corresponding reference on line 46.
Comments 4: Line 72: The authors should put the definition of PBNPs.
Response 4: Thank you for pointing it out. We agree with your point of view and define PBNPs in line 72.
Comments 5: Line 112: The authors should define IC50 the first time it is mentioned.
Response 5: Thank you for pointing it out. We agree with your viewpoint and define IC50 and IC10 in the abstract.
Comments 6: Line: despite DMSO being a common reagent, The authors should describe the definition.
Response 6: Thank you for pointing it out. We agree with your point of view and define DMSO on line 137.
Comments 7: Line 114: The authors should define ICTS.
Response 7: Thank you for pointing it out. We agree with your point of view and define ICTS in line 114.
Comments 8: Line 155: The authors should define BSA the first time it is mentioned as same as all the other abbreviations used in the paper.
Response 8: Thank you for pointing it out. We agree with your point of view and defined BSA when it was first mentioned.
Comments 9: Line 162: "10 mg of BSA was dissolved", The authors should correct the sentence to "were dissolved"
Response 9: Thank you for pointing it out. We agree with your point of view and have revised line 162 to 'were dissolved'.
Comments 10: Lines 153-160: The authors should rewrite this methodology in the past tense.
Response 10: Thank you for pointing it out. We agree with your point of view and have rewritten paragraph 2.7 in the past tense.
Comments 11:Line 188: the subtitle could be changed to "Evaluation of specificity" to make it more clear.
Response 11: Thank you for pointing it out. We agree with your viewpoint and have revised line 188 to 'Evaluation of specificity'.
Comments 12:Line 345: the spelling of "optimisation" is incorrect. Also, the term optimization involves the use of a design of experiment, which was not made in this work. It could be more appropriate to replace this word with another term, such as "comparison" or "evaluation".
Response 12: Thank you for pointing it out. We agree with your point of view and have made modifications to the words.
Round 2
Reviewer 2 Report
Comments and Suggestions for Authors
The authors considered the suggestions proposed.